# Surface Modification of Polyurethane Sponge with Zeolite and Zero-Valent Iron Promotes Short-Cut Nitrification

**DOI:** 10.3390/polym16111506

**Published:** 2024-05-26

**Authors:** Zexiang Liu, Yong Chen, Zhihong Xu, Jinxu Lei, Hua Lian, Jian Zhang, Zhiwei Wang

**Affiliations:** 1Guangxi Key Laboratory of Clean Pulp & Papermaking and Pollution Control, College of Light Industry and Food Engineering, Guangxi University, Nanning 530004, China; liuzexiang314@163.com (Z.L.); 2216391003@st.gxu.edu.cn (Y.C.); 2216301051@st.gxu.edu.cn (Z.X.); ricardododo@foxmail.com (J.L.); lihua321_wen@126.com (H.L.); 2Provincial and Ministerial Collaborative Innovation Center for Sugar Industry, Nanning 530004, China

**Keywords:** composite carrier, PN-A, nitrite accumulation, microbial adsorption, ammonia-rich microenvironment

## Abstract

Partial nitrification-Anammox (PN-A) is a cost-effective, environmentally friendly, and efficient method for removing ammonia (NH_4_^+^-N) pollutants from water. However, the limited accumulation of nitrite (NO_2_^−^-N) represents a bottleneck in the development of PN-A processes. To address this issue, this study developed a composite carrier loaded with nano zero-valent iron (nZVI) and zeolite to enhance NO_2_^−^-N accumulation during short-cut nitrification. The modified composite carrier revealed electropositive, hydrophilicity, and surface roughness. These surface characteristics correlate positively with the carrier’s total biomass adsorption capacity; the initial adsorption of microorganisms by the composite carrier was increased by 8.7 times. Zeolite endows the carrier with an NH_4_^+^-N adsorption capacity of 4.50 mg/g carrier. The entropy-driven ammonia adsorption process creates an ammonia-rich microenvironment on the surface of the carrier, providing effective inhibition of nitrite-oxidizing bacteria (NOB). In tests conducted with a moving bed biofilm reactor and a sequencing batch reactor, the composite carrier achieved a 95% NH_4_^+^-N removal efficiency, a NO_2_^−^-N accumulation efficiency of 78%, and a doubling in total nitrogen removal efficiency. This composite carrier enhances NO_2_^−^-N accumulation by preventing biomass washout, inhibiting NOB, and enriching PN-A functional bacteria, suggesting its potential for large-scale, stable PN-A applications.

## 1. Introduction

Ammonia (NH_4_^+^-N) is the primary nitrogenous pollutant released into the aquatic environment [1]. Its high accumulation can adversely affect both aquatic ecosystems [2] and human health [3]. Naturally, the amount of NH_4_^+^-N occurring in groundwater and surface water is below 0.2 mg/L [4]. The discharge of urban, industrial, and agricultural wastewater has caused a significant increase in NH_4_^+^-N concentrations in groundwater and surface water, with levels ranging from 100 mg/L to 1000 mg/L [5]. In marine environments, the safe level for NH_4_^+^-N is below 1 mg/L. China’s first-grade emission standard for NH_4_^+^-N in wastewater is set at less than 15 mg/L [6]. Consequently, the removal of nitrogenous pollutants from wastewater is essential. Anaerobic ammonia oxidation (anammox) has garnered considerable attention as an economical and environmentally friendly method for biological nitrogen removal, and it is used to treat ammonia-rich wastewater [7]. However, the anammox process requires the presence of both NH_4_^+^-N and nitrite (NO_2_^−^-N) in the feed water to occur [8]. To address this requirement, some researchers have proposed partial nitrification of NH_4_^+^-N to NO_2_^−^-N, which can then serve as a stable electron acceptor for the anammox reaction, a process known as partial nitrification-Anammox (PN-A) [9]. Short-cut nitrification is used to control the nitrification reaction in the NO_2_^−^-N stage, which can shorten the reaction process, reduce construction investment, and save the energy consumption from aeration.

The PN-A process is valued for its economic efficiency, energy-saving potential, and environmental benefits, but it is susceptible to several factors and typically has a lengthy start-up time [10]. Factors impacting the process include (1) nitrite-oxidizing bacteria (NOB) that can disrupt the accumulation of NO_2_^−^-N during the PN-A process, (2) reduced activity of anammox bacteria (AnAOB) due to oxygen aeration, and (3) loss of sludge in the reactor due to water flow. To expedite the start-up, it is crucial to maintain stable NO_2_^−^-N accumulation during the short-cut nitrification stage [11] while providing suitable conditions for AnAOB growth and ensuring high biomass retention, both of which are essential for the PN-A process [12].

NOB are less tolerant of free ammonia (FA) compared to ammonia-oxidizing bacteria (AOB), with NOB inhibition occurring at FA concentrations between 0.1 and 1 mg/L [13]. As a natural molecular sieve, zeolite exhibits effective ion adsorption and exchange characteristics, particularly for NH_4_^+^-N [14]. In the short-cut nitrification stage, the use of zeolite as a carrier with a high affinity for NH_4_^+^-N creates a microenvironment rich in ammonia, inhibiting NOB that are sensitive to it. In conjunction with microorganisms that utilize NH_4_^+^-N to form bio-zeolite, this microenvironment facilitates the desorption of adsorbed NH_4_^+^-N, providing a reaction substrate for these microorganisms. Zeolites, therefore, play a dual role as ion exchangers and biocarriers in the reactor [11,15,16]. In recent years, nano zero-valent iron (nZVI) has been found to significantly enhance the activity of AnAOB [17]. nZVI acts not only as a nutrient for AnAOB but also positively affects their growth [18] and activity [17,19]. The oxidation of nZVI results in localized anaerobiosis, creating a suitable growth environment for AnAOB [20].

Moving bed biofilm reactor (MBBR) technology is widely used in wastewater treatment for its efficiency as an attached growth treatment process [21]. Different types of biofilm carriers, such as polyethylene plastic, polyurethane sponge, and activated carbon, are commonly used in MBBR technology [22]. Among all types of carriers, polyurethane sponge carriers are considered ideal for microbial immobilization and attached-growth cultivation [21,23] due to their durability, chemical resistance [24], large specific surface area, high porosity [25], good fluidization properties [26], and biocompatibility [27]. However, the polyurethane sponge has a single composition, which limits the actual treatment efficiency of the MBBR. Thus, researchers have been focusing on modifying polyurethane carriers. They often create new carriers by directly adding target materials to the polyurethane matrix or by combining polyurethane with several materials of differing properties, providing the polyurethane carrier with the benefits of each component [27].

In this study, nZVI@Z-PU was prepared by loading zeolite and nZVI onto polyurethane sponge using an impregnation method. The nZVI@Z-PU was characterized by its surface properties, including roughness, hydrophilicity, and electropositive [28]; its ability to create an ammonia-rich environment; and its adsorption capability for activated sludge microorganisms. Additionally, moving bed biofilm reactor-sequencing batch reactor (MBBR-SBR) tests were conducted to monitor NH_4_^+^-N oxidation, NO_2_^−^-N accumulation, and total nitrogen removal to determine whether the material promotes highly selective nitrosation.

## 2. Materials and Methods

### 2.1. Material

#### 2.1.1. The Synthesis of nZVI

An improved liquid-phase reduction technique was adopted to prepare nZVI used in this work [29]. Initially, anhydrous FeCl_3_ (6 g) was dissolved in 50 mL of deoxygenated deionized water with PEG-4000 (2.5 g) as a dispersant. Liquid pH was adjusted to 4 with HCl before ultrasonic treatment for 10 min. Next, the mixture was transferred to a three-Neck flask and flushed with nitrogen. KBH_4_ (4.5 g) was dissolved in 20 mL deionized water and 30 mL ethanol. This solution was then added to the flask using a constant-pressure dropping funnel at a rate of 30 drops per minute. Throughout the reaction, a stirring rate of 400 rpm was maintained. After stirring for 1 h, the mixture was allowed to rest for 30 min to facilitate nZVI particle growth. The nZVI precipitate was collected by magnetic filtration and washed several times with deionized water and anhydrous ethanol under a stream of nitrogen. Subsequently, it was freeze-dried in a vacuum freeze-dryer for 24 h and stored in a vacuum-sealed container for further experimental research.

#### 2.1.2. Preparation of Nano Zero-Valent Iron@Zeolite Composite Carrier (nZVI@Z-PU)

A composite carrier was synthesized using the impregnation method. Zeolite powder (particle size < 10 μm) was mixed with nano zero-valent iron (nZVI) at a weight ratio of 8:1 and dispersed using ultrasonic dispersion for uniformity. A 10% (*w*/*v*) polyvinyl alcohol (PVA) solution was mixed with a 5% (*w*/*v*) polyquaternium-10 solution at a volume ratio of 10:1, then stirred for 20 min to ensure complete integration. The zeolite-nZVI powder, waterborne polyurethane, and PVA-polyquaternium-10 solution were mixed at a weight ratio of 27:40:4 to ensure the desired composition. The resulting mixture was stirred at 4 °C for 1 h to create a homogeneous solution. A number of 1 cm^3^ polyurethane sponge carriers were fully immersed in the homogeneous solution. After multiple rounds of impregnation and squeezing, the carriers were dried at 30 °C in a vacuum oven for 24 h to create a coating. Once coated, the carriers were washed three times with deionized water, dried, and then sealed in vacuum bags for storage. The process of composite carrier preparation is shown in Figure 1. Based on feedback from use in the laboratory, the composite carrier has a service life of at least 8 months.

#### 2.1.3. Production of Sludge Film

An amount of 5 mL of flocculent sludge was washed and thoroughly broken down, then transferred to a centrifuge tube, and distilled water was added to a total volume of 20 mL. The sample underwent centrifugation at 3000 rpm for 5 min, and the supernatant was discarded. The precipitate was resuspended in distilled water to a final volume of 20 mL. The resuspension was subsequently passed through a 0.45 μm aqueous filter membrane, which was washed three times with deionized water. The filter membrane was placed on an agar plate to moisten, then dried at room temperature for 4 h before the experiment.

### 2.2. Methods

#### 2.2.1. Composite Carrier Characterization

Due to the volume and morphology of the composite carrier, characterization is limited. To ensure effective characterization, the composite carrier was frozen in liquid nitrogen and then ground in a dry agate mortar, which had been scrubbed with alcohol, before being passed through a 200-mesh sieve.

G2F20U-TWIN transmission electron microscopy (FEI Company, Hillsboro, OR, USA) was used to observe the size and micro-morphology of the prepared nZVI at an operating voltage of approximately 100 kV, and SU8020 field emission scanning electron microscopy (Hitachi High Technologies Corporation, Tokyo, Japan) was used for morphological analysis of the prepared composite carriers. The composite carriers were analyzed using ESCALAB250XI + X-ray photoelectron spectroscopy (Thermo Fisher Scientific, Waltham, MA, USA). The chemical composition of the surface was primarily determined through full-spectrum scanning, while the chemical state and structure of specific elements were examined through narrow-Area scanning. The composites were analyzed by D/MAX 2500V X-ray diffractometer (Rigaku Corporation, Tokyo, Japan) for physical phase and crystallographic analysis at a scanning rate of 2 degrees per minute, with a scanning range of 5 to 90 degrees, operating at 40 kV and 40 mA. TENSOR II fourier transform infrared spectrometer (Bruker Corporation, Billerica, MA, USA) was used to detect the active functional groups of the materials in the range of 250–4000 cm^−1^ with a resolution of 4 cm^−1^. The Zeta potential of the milled and sieved carrier powder and the crushed and washed flocculent sludge was determined using a Nano-ZS90X Zeta potential analyzer (Malvern Instruments, Malvern, UK). The contact angle between the carrier, sludge film, and water was determined using a DSA100E contact angle meter (Bruker Corporation, Billerica, MA, USA).

#### 2.2.2. Static Adsorption of Bacteriophages by Composite Carriers

The activated sludge was washed 3 times with 0.1 M phosphate-buffered saline (PBS). After the supernatant was removed, 5 g of activated sludge was dispersed in 50 mL of sterile water and vortexed for 5 min. The mixture was then centrifuged at 800 rpm for 5 min, after which the supernatant was discarded. An amount of 1 mL of the sediment was added to 100 mL of sterile water to create a 1 g/L bacterial suspension.

An amount of 10 mL of the bacterial suspension was added to a 500 mL serum bottle containing 200 mL of nutrient matrix consisting of 100 mg-N/L NH_4_Cl and 100 mg-N/L NaNO_2_ as nitrogen sources, 40 mg/L KH_2_PO_4_, 800 mg/L NaHCO_3_, 36 mg/L CaCl_2_-2H_2_O, 25 mg/L MgCl_2_-6H_2_O, and 1 mL/L trace elements. A total of 10 composite carriers and 10 original carriers were added, and the serum bottles were placed in a water bath shaker set to 120 rpm and 30 °C. Three parallel groups were set up for each reaction. After 24 h of adsorption, the carriers were removed with sterile tweezers, the surfaces were rinsed with sterilized water 3 times and then fixed in 2.5% glutaraldehyde at 4 °C for 1–3 h. The carriers were then rinsed with sterile 0.01 M PBS 3 times and subjected to gradient dehydration in 10%, 30%, 50%, 70%, and 90% ethanol, each for 10 min in sequence. The carriers were then dehydrated in anhydrous ethanol for 30 min [30], before being freeze-dried for 6 h. After drying, the carrier was processed into a 5 mm square sample with a height of about 3 mm. The surface was coated with gold and then analyzed in an SEM vacuum chamber at an accelerating voltage of 3 kV to observe the surface characteristics.

After the adsorption experiment, 3 composite carriers and the original carriers were taken out and repeatedly washed with sterile water. Carriers that did not undergo adsorption were used as the control group. The amount of bacterial gel clusters on the surface of the carriers was determined by the crystal violet staining method. The cleaned samples were dried at 40 °C for 45 min, then immersed in a 1% crystal violet solution for 45 min. After staining, the samples were taken out, washed three times with sterile water, and then left to air dry for 6 h. The dried samples were placed in 30 mL centrifuge tubes filled with 95% ethanol solution, then vortexed for 3 min, and subsequently ultrasonicated for 10 min to detach the stained bacterial clusters from the carrier. The absorbance of the detached solution was measured at 595 nm using a UV-visible spectrophotometer. The total amount of bacterial colloid was inferred from the absorbance at 595 nm.

#### 2.2.3. Composite Carrier Ammonia Adsorption Experiment

The adsorption experiments were carried out by static oscillation adsorption. A 200 mL NH_4_^+^-N solution with a concentration of 100 mg/L was prepared in a 500 mL conical flask, and 20 piece composite carriers and 20 piece original carriers were added separately. The pH was adjusted to 7, and the solution was shaken with a water bath oscillator at 120 r/min for 6 h at 25 °C. Water samples (2 mL) were taken at regular intervals and filtered through 0.45 μm polyethersulfone membranes. The absorbance from the water samples containing the original carriers was measured spectrophotometrically as a blank group. NH_4_^+^-N and NO_2_^−^-N were analyzed using ultraviolet spectrophotometry, while nitrate (NO_3_^−^-N) was analyzed using 2-isopropyl-5-methyphenol spectrophotometry, with test methods based on the Chinese standard methods for the examination of water and wastewater (China, 2019). The temperature was changed under the same experimental conditions, and the concentration of NH_4_^+^-N at which the adsorption reached equilibrium was recorded as T = 35 °C and T = 45 °C. Three trials were performed to achieve a standard error of less than 5%.

(1)NH_4_^+^-N adsorption ($q_{t}$) and NH_4_^+^-N removal in solution (η) for individual carriers
(1)$$q_{t}=\frac{(c_{0}-c_{t})V}{M}$$
(2)$$\eta =\frac{c_{0}-c_{t}}{c_{0}}\times 100\%$$
where *q_t_* is NH_4_^+^-N adsorbed on adsorbent (zeolite loaded on carrier) (mg/g carrier); *c*_0_ is the initial concentration of NH_4_^+^-N solution (mg/L); *c_t_* is the concentration of NH_4_^+^-N solution at time t (mg/L); *V* is the volume of NH_4_^+^-N solution (L); and *M* is mass of adsorbent (the average weight of a single composite carrier is 0.124 g).(2)To determine the adsorption isotherm model for NH_4_^+^-N adsorption on the carrier, the modified Langmuir isotherm model and the Freundlich adsorption isotherm model were fitted.Langmuir adsorption isotherm model
(3)$$\frac{1}{q_{e}}=\frac{1}{q_{m}K_{l}}\times \frac{1}{c_{e}}+\frac{1}{q_{m}}$$Freundlich adsorption isotherm model
(4)$$\mathrm{lg}q_{e}=\frac{1}{n}\mathrm{lg}c_{e}+\mathrm{lg}K_{F}$$
where *q_e_* is equilibrium adsorption capacity (mg/g carrier); *c_e_* is the equilibrium concentration of NH_4_^+^-N in solution (mg/L); *q_m_* is saturated adsorption capacity (mg/g carrier); *K_l_* is the adsorption constant; *K_F_* is the adsorption coefficient, indicating the adsorption capacity of the adsorbent; and *n* is a constant, usually greater than 1.(3)In order to quantify the relationship between reaction rate and NH_4_^+^-N concentration in the adsorption reaction, an adsorption kinetic model was fitted.Pseudo-first-order model
(5)$$\frac{dq}{dt}=K_{1}(q_{e}-q)$$It can be morphed into:(6)$$\ln(q_{e}-q_{t})=\ln q_{e}-K_{1}t$$Pseudo-second-order model
(7)$$\frac{dq}{dt}=K_{2}{\left(q_{e}-q\right)}^{2}$$It can be morphed into:(8)$$\frac{t}{q_{t}}=\frac{1}{{{K_{2}q}_{e}}^{2}}+\frac{t}{q_{e}}$$
where *q_t_* is NH_4_^+^-N adsorption capacity of the adsorbent at time t (mg/g carrier) and *K*_1_ is the equilibrium constant. *K*_2_ is the equilibrium constant.(4)Thermodynamic modeling of adsorption reactions in order to clarify whether the reaction can proceed spontaneously and how it is driven by the reaction
(9)$$\ln c_{e}=-\ln K+\frac{△H}{RT}$$In case of ideal adsorption:(10)$$△G=-RTlnK_{C}=△H-T△S$$It can be morphed into:(11)$$lnK_{C}=-\frac{△H}{RT}+\frac{△S}{R}$$
(12)$$K_{C}=\frac{c_{0}-c_{e}}{c_{e}}\times \frac{V}{M}$$
where *c_e_* is the equilibrium concentration of NH_4_^+^-N in solution (mg/L); *R* is the molar gas constant (8.314 J·mol^−1^·K^−1^); *T* is the solution temperature (K); *K* is the equilibrium constant related to a chemical reaction or process; *K_c_* is the equilibrium adsorption partition coefficient; Δ*G* is the Gibbs free energy (J·mol^−1^); Δ*H* is the enthalpy change (KJ·mol^−1^); Δ*S* is the entropy change (J·mol^−1^·K^−1^); *c*_0_ is the initial concentration of NH_4_^+^-N solution (mg/L); *M* is the mass of adsorbent (g); and *V* is the volume of adsorbed solution (L).

#### 2.2.4. Short-Cut Nitrification Experiments

The experiments were conducted in a laboratory-scale SBR with a complete cycle consisting of 5 min for influent, 335 min for reaction, 10 min for settling, and 10 min for effluent, with a drainage ratio of 60% and a working volume of 1.7 L. A mixture of nitrifying activated sludge (150 mL, from the laboratory’s nitrification reactor) and anammox-enriched flocs (50 mL, from the laboratory’s sealed and preserved anammox sludge) was inoculated into the reactor, resulting in an initial biomass concentration of about 1.5 g/L of volatile suspended solids (VSS). The reactor was then filled with carriers at a rate of approximately 10%. The source of nitrogen for the synthetic wastewater was NH_4_^+^-N (with the addition of NH_4_Cl) at a concentration of 110 ± 2.3 mg/L. Other components and trace elements in the synthetic wastewater were prepared according to [31]. The pH in the reactor was controlled at 7.5–8.23 by adding NaHCO_3_ during continuous operation. Oxygen was supplied by a gas pump through a microbubble aerator located at the bottom of the reactor. The aeration rate was regulated by a gas flow meter with an airflow rate of 0.2 L/min, and the temperature in the reactor was maintained at 35 ± 1 °C using a temperature-controlled water bath throughout the experiments.

The total nitrogen removal efficiency (TNRE), nitrite accumulation efficiency (NAE), ammonia removal efficiency (ARE), ammonium removal rate (ARR), ammonia oxidation rate (AOR), nitrite oxidation rate (NOR), and anammox nitrogen removal rate by anammox (ANRR) for the SBR reactor were calculated as shown in Equations (13) to (19), respectively.
(13)$$TNRE(\%)=\frac{C{\left(TN\right)}_{inf}-{C(TN)}_{eff}}{{C(TN)}_{inf}}\times 100$$
(14)$$NAE(\%)=\frac{{{{C(NO}_{2}}^{-}-N)}_{eff}}{C{{{(NO}_{2}}^{-}-N)}_{eff}+C({{{NO}_{3}}^{-}-N)}_{eff}}\times 100$$
(15)$$ARE(\%)=\frac{C({{{NH}_{4}}^{+}-N)}_{inf}-C({{{NH}_{4}}^{+}-N)}_{eff}}{C{{{(NH}_{4}}^{+}-N)}_{eff}}\times 100$$
(16)$$ARR(kg-N/m^{3}\cdot d^{-1})=\frac{C({{{NH}_{4}}^{+}-N)}_{inf}-C({{{NH}_{4}}^{+}-N)}_{eff}}{HRT}\times 24\times 10^{-3}$$
(17)$$AOR(kg-N/m^{3}\cdot d^{-1})=\frac{C({{{NH}_{4}}^{+}-N)}_{inf}-C({{{NH}_{4}}^{+}-N)}_{eff}-\frac{△C(TN)}{2.04}}{HRT}\times 24\times 10^{-3}$$
(18)$$NOR(kg-N/m^{3}\cdot d^{-1})=\frac{C({{{NO}_{3}}^{-}-N)}_{eff}-\frac{0.26△C(TN)}{2.04}}{HRT}\times 24\times 10^{-3}$$
(19)$$ANRR(kg-N/m^{3}\cdot d^{-1})=\frac{△C(TN)}{HRT}\times 24\times 10^{-3}$$
where subscript “*inf*” refers to influent; subscript “*eff*” refers to effluent; *HRT* stands for hydraulic retention time (h); and *TN* denotes total nitrogen.

## 3. Results and Discussion

### 3.1. Effect of Modification on the Surface Morphology and Properties of Carriers

#### 3.1.1. Morphological Changes in Composite Carriers

As can be seen from Figure 2a, the prepared black particles are capable of magnetic separation and possess ferromagnetism, which is initially inferred to be zero-valent iron [29]. The morphology of the material was investigated using TEM.

From Figure 2b, it can be observed that the prepared material has particle sizes ranging from 50 to 110 nm, with a nanospherical structure and a high tendency to agglomerate. Additionally, the nanoparticles exhibit a chain-like aggregation structure due to inherent magnetic and electrostatic interactions [32]. Figure 2c shows that individual particles consist of a dense core surrounded by a thin shell, and further magnification reveals a distinctive core-shell structure. This characteristic is due to the highly reactive chemical nature of nZVI, where the outer iron atoms are prone to oxidation. The core–shell structure comprises an iron core and an outer passivated shell of iron oxide, which is a characteristic feature of nZVI [33,34]. The powder material prepared by the liquid-phase reduction method has a shell-like structure of ferromagnetic nanoparticles on its surface, indicating successful synthesis of nZVI.

Figure 3 illustrates the SEM images of the carrier. The sponge carrier (Figure 3a) exhibits a porous frame structure, whereas the polyurethane framework of the unmodified carrier appears relatively smooth, without additional structural branches (Figure 3b). In contrast, the composite carrier (Figure 3c,d), due to the incorporation of zeolite and nZVI, exhibits increased surface roughness. In the EDS images of the carrier, the distribution of C, N, and O elements in the original carrier aligns with the polyurethane framework, while Al, Si, and Fe are quite scarce. However, in the composite carrier, due to the successful incorporation of zeolite and nZVI, the levels of Al, Si, and Fe are significantly higher than in the original carrier, with their distribution corresponding to that of the polyurethane skeleton. Spherical nanoparticles and zeolite particles of similar size are uniformly dispersed across the carrier’s surface and within its crevices. From the detailed images of the composite carrier (Figure 3d-1 to Figure 3d-4), it is evident that the distribution of nZVI aligns closely with that of the zeolite, with nZVI covering the zeolite without any observed aggregation. This uniform distribution is achieved by thoroughly mixing zeolite and nZVI during the synthesis of the composite carrier. During this process, some nZVI becomes dispersed and embedded within the pore structure of the zeolite. By serving as a dispersing agent, zeolite’s spatial structure and lattice changes mitigate the aggregation of nZVI, protecting the embedded nZVI from rapid oxidation [34,35].

Zeolite and nZVI were successfully added to polyurethane sponge carriers through a simple and controllable impregnation method. The zeolite aided in dispersing nZVI, preventing its agglomeration. The uniform distribution of zeolite and nZVI on the polyurethane sponge increased the carrier’s surface roughness. This rougher surface aids in resisting hydraulic shear during the initial stage of microbial adhesion, promoting microbial colonization, and leading to faster biofilm formation [27].

#### 3.1.2. Elemental and Structural Analyses of Composite Carriers and Carrier Surface Properties before and after Modification

Figure 4a shows the XRD pattern, where a distinct peak at 2θ = 44.5° indicates the presence of Fe^0^, confirming the successful synthesis of nZVI [36]. The characteristic peaks for zeolite and nZVI on the composite carrier suggest that they were successfully loaded onto the composite carrier. However, due to polyurethane interference, a slight shift in the characteristic peaks for zeolite and nZVI was observed in the composite carrier’s XRD pattern. To further investigate whether the nZVI changed the carrier synthesis, XPS was used to analyze the elemental states and chemical composition of the composite carrier, as shown in Figure 4b–f. In the C 1s spectrum, peaks with binding energies at 284.8 eV, 286.4 eV, 288.7 eV, and 298.0 eV correspond to C-C, C-O, C=O, and O=C-N functional groups, respectively [37,38]. The Fe 2p spectrum shows a binding energy peak for Fe^0^ at 706.92 eV, confirming that nZVI retains its zero-valent iron characteristic when loaded onto the composite carrier. Additionally, the Fe_2_O_3_ (712.3 eV and 725.8 eV) and Fe_3_O_4_ (710.1 eV and 723.8 eV) peaks indicate the presence of iron oxides on the composite carrier’s surface, stemming from the core-shell structure of nZVI [33]. In the Al 2p spectrum, peaks at 74.3 eV and 74.8 eV align with the Si 2p peaks at 102.0 eV and 102.5 eV, indicating the characteristic structure of the zeolite, which mainly consists of microporous hydrated aluminosilicates derived from aluminum and silicon [39]. Binding energy peaks for Al_2_O_3_ (75.9 eV) and SiO_2_ (103.8 eV) from the zeolite components were also observed in the Al 2p and Si 2p spectra.

During preparation, the incorporation of the cationic polymer polyquaternium-10 increased the surface zeta potential of the composite carrier from -7.45 mV to 1.79 mV (Figure 4g). The sludge membrane, in contrast, consistently exhibited an electronegativity, in line with the typical electronegativity of most microorganisms, stemming from the presence of surface anions [40]. The electropositive surface of the composite carrier is anticipated to strengthen the electrostatic attraction between the carrier and the microorganisms [20]. Figure 4(i-1,i-2) illustrate the contact angle of the carrier with water before and after modification. After modification, the contact angle changed from a hydrophobic 113° to a completely hydrophilic 0°. This change is ascribed to the inclusion of PVA and zeolite, both hydrophilic materials, during the fabrication process [20]. During the preparation of the composite carrier, waterborne polyurethane, with the same composition as the polyurethane sponge, was used as the binding agent. The surface of the zeolite was coated with a thin polyurethane film, which might impact the NH_4_^+^-N adsorption capability of the zeolite on the composite carrier. Figure 4h reveals a distinct characteristic peak for NH_4_^+^-N at 1450 cm^−1^ [41]. The polyurethane film’s coverage did not impact the NH_4_^+^-N adsorption of the composite carrier, indicating that the waterborne polyurethane hybrid film on the surface of the composite carrier is not impermeable. It contains small pores, allowing the exchange of substances of specific sizes with those within the film.

The nZVI retained its zero-valent iron properties during carrier loading, likely due to the dispersion effect of zeolite, which prevented nZVI oxidation. Surface modification of the carrier imparts a positive zeta potential and hydrophilicity. Furthermore, the loading process did not impede the zeolite’s ammonia adsorption capability, allowing the carriers to function effectively as biocarriers in a PN-A reactor.

### 3.2. Static Adsorption of Microorganisms before and after Carrier Modification

The initial adsorption of microorganisms by both the original carrier and the composite carrier after 24 h of microbial adsorption experiments is illustrated in Figure 5a–c. The composite carrier demonstrated significantly higher initial adsorption compared to the original carrier. The initial adsorption of microorganisms by the carrier was semi-quantitatively determined by staining with crystal violet [30], and the adsorption of microorganisms by the composite carrier was 8.7 times higher than that of the original carrier.

A hydrophilic and electropositive carrier surface can enhance biofilm formation [20]. Due to the addition of hydrophilic zeolite, PVA, and the cationic surfactant polyquaternium-10 during the preparation process, the composite carrier exhibits excellent hydrophilicity and electropositive. These modifications significantly improve the carrier’s ability to adsorb microorganisms. Additionally, the loading of zeolite and nZVI increases the surface roughness of the composite carrier, providing sites for microorganisms to attach and hide, helping them resist hydraulic shear in the reactor. This rougher surface ensures that microorganisms are less likely to be washed out of the reactor; microorganisms nested in the rough surface grooves of the carrier resisted hydrodynamic scouring. These changes in the composite carrier’s surface properties enhance the initial microbial adhesion, which is the “rate-controlling step” in biofilm formation [42]. As a result, the composite carrier offers superior initial adhesion capability compared to the original carrier.

Microbial retention leads to the accumulation of microorganisms on the carrier, which helps prevent the loss of functional bacteria. Sufficient microbial abundance is a critical prerequisite for removing nitrogen pollutants from water and achieving partial nitrification.

### 3.3. Research on the Model and Mechanism of NH_4_^+^-N Adsorption on the Composite Carrier

NH_4_^+^-N removal before and after carrier modification is shown in Figure 6a, and the adsorption capacity of NH_4_^+^-N by composite carriers is shown in Figure 6b. The removal of NH_4_^+^-N by the composite carrier exhibits a trend of initial increase followed by stabilization, a pattern observed across various temperatures. The composite carrier rapidly adsorbs NH_4_^+^-N during the initial reaction phase, but the adsorption rate gradually plateaus over time. This trend can be attributed to the ion-exchange process where zeolite’s metal cation sites are progressively replaced by NH_4_^+^-N, leading to saturation of adsorption in later stages. However, an increase in temperature promotes the migration of NH_4_^+^-N into the zeolite pores, resulting in a higher removal rate and adsorption capacity [43]. The original carrier demonstrated almost no capacity for NH_4_^+^-N adsorption, exhibiting virtually zero ability to adsorb NH_4_^+^-N. In contrast, the composite carrier displayed a dynamic pattern of NH_4_^+^-N adsorption. At 35 °C, the composite carrier achieved a removal rate of 50% within the first hour of the reaction. Figure 6b also shows that the NH_4_^+^-N adsorption capacity of the carriers increases rapidly during the first hour, indicating significant NH_4_^+^-N adsorption. However, the adsorption rate gradually slows down afterward, reaching a maximum adsorption capacity of 4.50 mg/g carrier after 8 h.

The adsorption mechanism of NH_4_^+^-N by the composite carrier at 35 °C was illustrated in Figure 6c–f. The Langmuir equation model presented better fitness (R^2^ = 0.99303) to NH_4_^+^-N adsorption than the Freundlich equation model (R^2^ = 0.92862), suggesting that the adsorption followed a chemisorption process on a monomolecular layer. In terms of kinetics, the pseudo-second-order model was a better fit (R^2^ = 0.9968), indicating that the adsorption of NH_4_^+^-N by the composite carrier begins rapidly and then slows, allowing for high-capacity NH_4_^+^-N adsorption in a shorter time. Thermodynamic analysis revealed that the change in adsorption-free energy (Δ*G* < 0), enthalpy (Δ*H* > 0), and entropy (Δ*S* > 0) indicates that the process of NH_4_^+^-N adsorption on the composite carrier is an entropy-driven, spontaneous heat-Absorbing reaction [44]. As the external NH_4_^+^-N concentration decreases, NH_4_^+^-N is released from the carrier, maintaining a higher NH_4_^+^-N concentration in the environment. This forms an ammonia-rich microenvironment on the surface of the carrier, which is both stable and continuous. The presence of NOB can disrupt short-cut nitrification, while the ammonia-rich conditions help suppress NOB activity and simultaneously provide the necessary substrates for the enrichment of AOB and AnAOB [45]. This enables the efficient implementation of the PN-A process.

### 3.4. Effect of the Composite Carriers on the Nitrosation Process

The nitrogen removal performance is shown in Figure 7. Initially, an unusual phenomenon was observed in that the effluent NH_4_^+^-N concentration from the composite carrier increased from 40.69 mg/L to 73.23 mg/L. This increase can be attributed to the initial adsorption of a large amount of NH_4_^+^-N by the composite carrier. As adsorption reached saturation, the carrier’s ability to adsorb NH_4_^+^-N diminished, resulting in higher effluent NH_4_^+^-N levels. This adsorption–desorption equilibrium created an ammonia-rich microenvironment on the composite carrier’s surface. The ammonia oxidation rate (AOR), nitrite oxidation rate (NOR), and ammonia–nitrogen removal rate (ANRR) can indirectly reflect the in situ activity of corresponding nitrifying microorganisms [46]. Figure 7e shows an increase in AOR for both reactors, suggesting that aeration increased the activity of AOB, resulting in a continuous decline in effluent NH_4_^+^-N for both the original and composite carriers. As shown in Figure 7c,d, the NH_4_^+^-N removal efficiency for both the original and composite carriers exceeded 95%.

Regarding NO_2_^−^-N and NO_3_^−^-N removal, the composite carrier’s NO_2_^−^-N levels stabilized at approximately 71.50 mg/L after peaking on day 10, with NO_3_^−^-N levels stabilizing at around 21.30 mg/L. In contrast, with the original carrier, NO_2_^−^-N levels declined after peaking, while NO_3_^−^-N levels continued to increase. By the end of the 20-day reaction, NO_2_^−^-N accumulation in the original carrier had decreased to 40%, while in the composite carrier it remained stable at around 78%, indicating that the original carrier achieved less effective short-cut nitrification compared to the composite carrier. The reason for this difference is that the composite carrier can adsorb up to 4.50 mg/g of NH_4_^+^-N, providing excellent buffering capacity and maintaining a high FA level. The ammonia-rich microenvironment created by the composite carrier inhibits NOB activity [11], resulting in stable NO_2_^−^-N accumulation and enabling short-cut nitrification. In contrast, the original carrier showed limited inhibition of NOB during the reaction, allowing accumulated NO_2_^−^-N to be converted to NO_3_^−^-N.

In Figure 7f, the ANRR for the composite carrier was consistently higher than that for the original carrier. AnAOB can be inhibited during intensive aeration, but, according to Liu et al. (2021), nZVI can create localized anaerobic conditions through its oxidation while also producing the iron ions necessary for AnAOB metabolism [20]. This is crucial for the growth of AnAOB. Additionally, the composite carrier’s adsorption–desorption cycle of NH_4_^+^-N provides a continuous supply of NH_4_^+^-N as a reaction substrate for AnAOB within the biofilm [47]. The composite carrier’s enhanced initial microbial adsorption capability could lead to a higher microorganism count compared to the original carrier. These factors contribute to the enhanced activity of AnAOB on the composite carrier. This increased microbial activity on the composite carrier might explain its superior performance in removing nitrogen pollutants from water while maintaining stable NO_2_^−^-N levels.

Sufficient nitrite accumulation and favorable AnAOB activity are prerequisites for achieving PN-A. Additionally, this composite carrier suppresses non-target NOB strains, which enhances the selectivity of functional microbes, thereby providing suitable conditions for a stable and efficient PN-A process.

## 4. Conclusions

In this study, a polyurethane composite carrier loaded with nZVI and zeolite was developed. Unlike traditional polyurethane carriers, which exhibit electropositivity, hydrophilicity, and higher surface roughness, these modifications significantly enhance microbial adhesion. The initial adsorption of microorganisms increased by 8.7 times. The composite carrier also established a sustained ammonia-rich microenvironment, which inhibits NOB. In reactors filled with this composite, a 95% removal efficiency of NH_4_^+^-N and a 78% accumulation efficiency of NO_2_^−^-N was achieved, demonstrating excellent short-cut nitrification performance. Additionally, the composite carrier’s TN removal rate was double that of a polyurethane carrier, indicating a higher level of AnAOB activity. This study shows that the composite carrier effectively prevents microbial washout, enhances NO_2_^−^-N accumulation, eliminates NOB in situ, and maintains a high level of activity for both AOB and AnAOB. As a result, the system leads to a stable and efficient short-cut nitrification process, providing favorable conditions for subsequent anammox treatment, which is crucial for the large-scale application of the PN-A process.

## Figures and Tables

**Figure 1 polymers-16-01506-f001:**
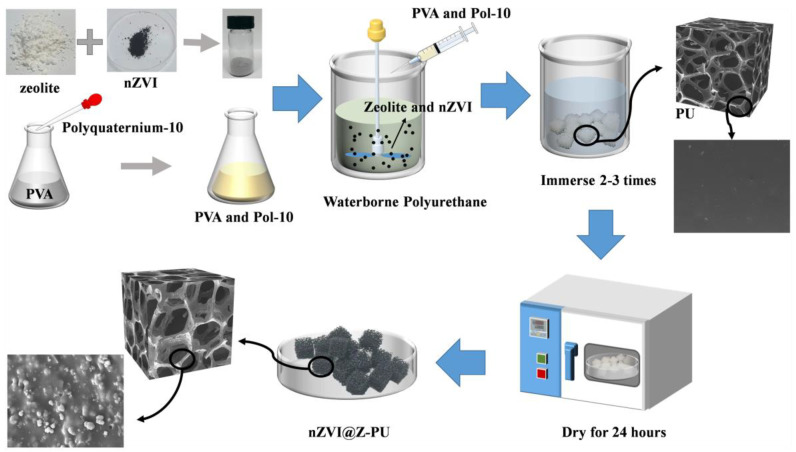
Flowchart for the preparation of the nano zero-valent iron@zeolite composite carrier.

**Figure 2 polymers-16-01506-f002:**
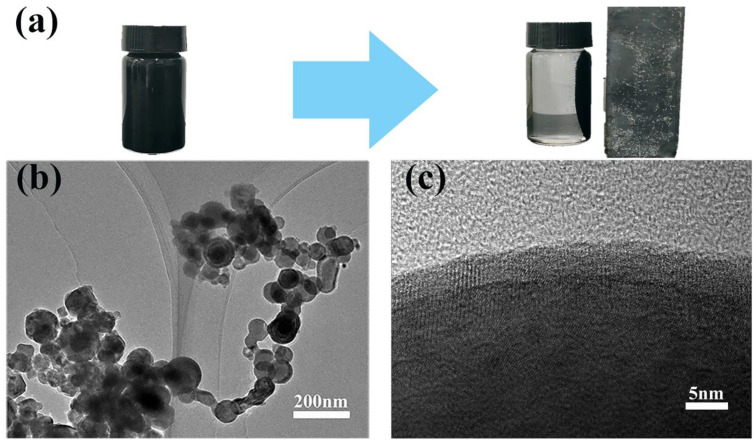
Ferromagnetism and TEM images of nZVI: (**a**) ferromagnetism of nZVI; (**b**) nZVI agglomerated in chain-like structures; and (**c**) core-shell structure of a single nZVI particle.

**Figure 3 polymers-16-01506-f003:**
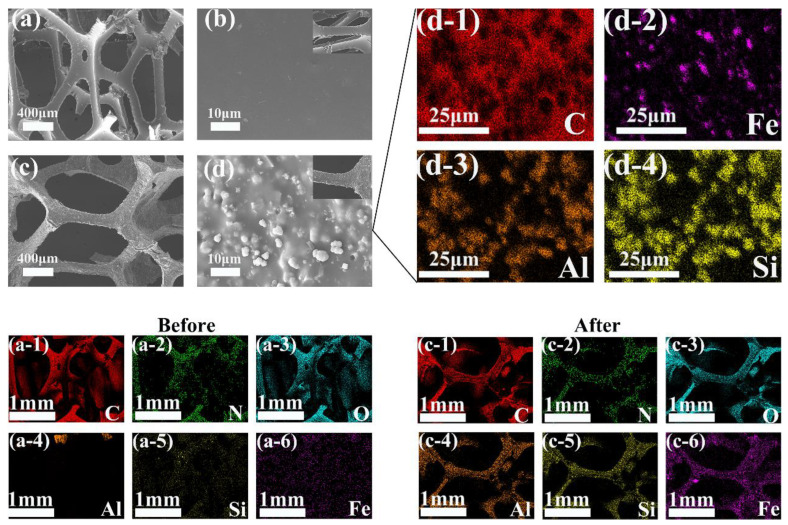
Morphological changes before and after carrier modification: (**a**–**d**) morphological changes and local detail images of carriers before and after modification; (**a-1**–**a-6**) elemental distributions of carriers before modification; (**c-1**–**c-6**) elemental distributions of carriers after modification; and (**d-1**–**d-4**) elemental distributions of local details of carriers after modification.

**Figure 4 polymers-16-01506-f004:**
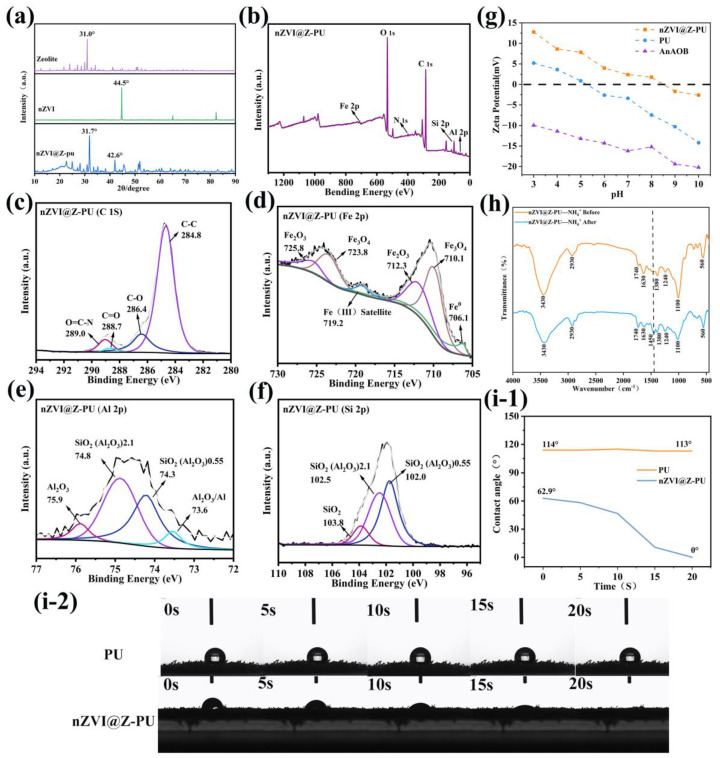
Characteristic changes in the carrier before and after modification: (**a**) composition of the carrier after modification, as determined by XRD; (**b**–**f**) elemental states and chemical composition of the carrier after modification, as revealed by XPS; (**g**) increase in surface zeta potential resulting from carrier modification; (**h**) FT-IR differences in the composite carrier after NH_4_^+^-N adsorption; and (**i-1**,**i-2**) improvement in hydrophilicity due to carrier modification.

**Figure 5 polymers-16-01506-f005:**
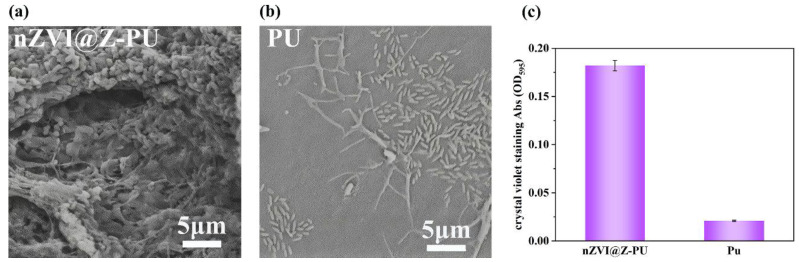
The comparison of initial microbial adsorption before and after carrier modification: (**a**) adsorption of microorganisms on carriers after modification; (**b**) adsorption of microorganisms on carriers before modification; and (**c**) measurement of microbial adsorption on carriers before and after modification using the crystal violet staining method.

**Figure 6 polymers-16-01506-f006:**
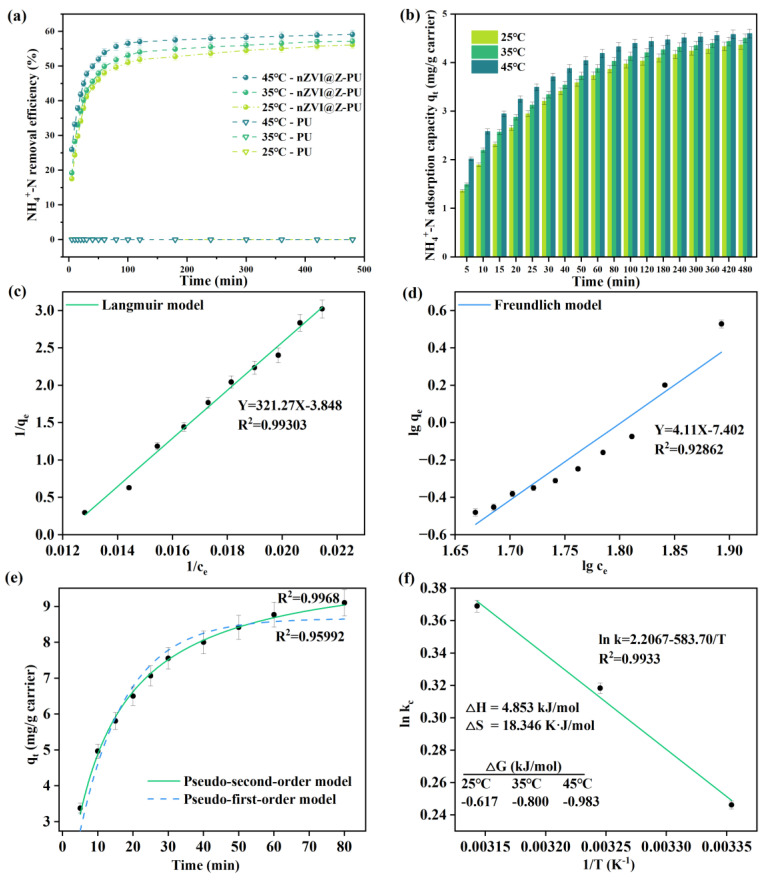
Kinetics of NH_4_^+^-N adsorption by the composite carrier at 35 °C: (**a**) NH_4_^+^-N removal quantity before and after carrier modification; (**b**) adsorption capacity of the modified carrier for NH_4_^+^-N; (**c**,**d**) isotherm adsorption model fitting; (**e**) kinetic equation fitting; and (**f**) thermodynamic equation fitting.

**Figure 7 polymers-16-01506-f007:**
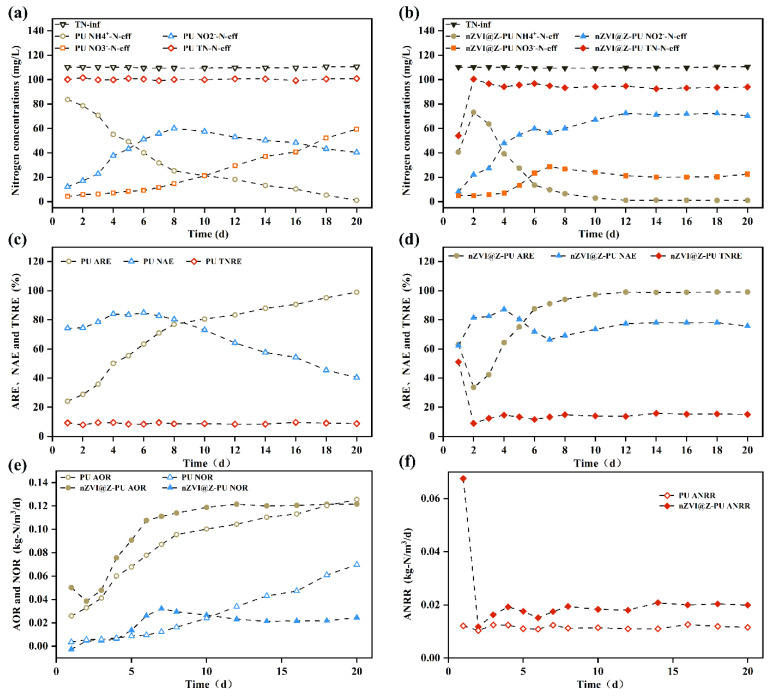
Changes in nitrogenous pollutants in experimental effluent water: (**a**,**b**) effects of carrier modification on TN, NH_4_^+^-N, NO_2_^−^-N, and NO_3_^−^-N in experimental effluent before and after modification; (**c**,**d**) effects of carrier modification on ammonia removal efficiency (ARE), nitrite accumulation efficiency (NAE), and total nitrogen removal efficiency (TNRE); and (**e**,**f**) effects of carrier modification on ammonia oxidation rate (AOR), nitrite oxidation rate (NOR), and nitrogen removal rate by anammox (ANRR).

## Data Availability

Data are contained within the article.

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
