# Peer review of "Surface Modification of Polyurethane Sponge with Zeolite and Zero-Valent Iron Promotes Short-Cut Nitrification"

_polymers, 2024, doi:10.3390/polym16111506_

Round 1
Reviewer 1 Report
Comments and Suggestions for Authors
The authors have carried out an interesting research work but the manuscript needs the thorough revision before it could be published in polymers.
Author must address the following queries.
1. Reframe the abstract with significant information and better avoid the long sentences in abstract "Additionally, the loading of zeolite enabled the composite carrier to exhibit rapid ammonia adsorption, endowing the carrier with an ammonia adsorption capacity of 4.5 mg/g carrer, and the adsorption of ammonia was an entropy-driven process, which could form an ammonia-rich microenvironment on the surface of the carrier, and consequently, inhibit the nitrite-oxidizing bacterium."
2. In abstract "an ammonia adsorption capacity of 4.5 mg/g carrer" is it correct? recheck the sentence.
3. Check the unit and typo errors throughout the manuscript.
4. In results and discussion after each subsection the significance of that study must be discussed.
5. Rewrite the conclusion section by discussing the significant results with numbers and also application and future scope of the research work.
6. Check the references for relevance and recent research works.
Comments on the Quality of English LanguageThe manuscript needs the thorough revision before it could be published in polymers.
Author Response
We sincerely appreciate your valuable feedback, which we use to improve the quality of the manuscript. Responses to your comments can be found in the attachment. Changes/additions to the manuscript are given in red text.

Reviewer 2 Report
Comments and Suggestions for Authors
Authors must address the following points before being considered for publication:
1. Should the authors justify in the introduction why they selected polyurethane for this application?
2. The authors must indicate the frequently found values of ammonia in water, maximum permitted levels, etc.
3. The authors should explain in methodology, more details of how they controlled the exact amount of zeolite anchoring on polyurethane, as well as the degree of quaternization of the surface.
4. The authors must include an illustrative figure about the mechanism of action of this new carrier and its processes involved with the chemical species of interest.
5. In the conclusion section, correct the writing of the chemical formulas, review the entire manuscript to avoid any errors.
6. The authors must explain the influence of water salts on the performance of this new material.
7. The authors should explain the life cycle of this new modified surface, how many cycles can it work continuously?
8. A process diagram could be of great help to readers
Author Response

(The authors gave the same response as above.)

Reviewer 3 Report
Comments and Suggestions for Authors
A composite carrier was synthesized by incorporation of zeolite and nanoscale zero-valent iron onto a polyurethane foam substrate for short-cut nitrification. The authors characterized this composite by various analytical methods and evaluated this composite performance. Logic is acceptable, but their results are expectable, and its performance is moderate, I think. (As I am not a specialist in this field, and thus I just compared this paper performance to those of other related paper.). In addition, in the respect of the materials, I could not find any originality and new concept in this field. Thus, I cannot strongly recommend the publication of their paper. Furthermore, their paper lacks the correctness, adequate explanation, and frequently contains poor English sentence, which make it difficult for the readers to understand their paper.
I really hope my following comments are useful for them to revise their paper in the next submission.
(1) I recommend them to ask native English speakers to check their paper because their manuscript sometimes is hard to read and understood.
For example, Line 161-163. Etc.
(2) In addition to (1), they should be careful of sentences. For example, line 55 (compare), 59 (the), 61 (And Together), 62 (utilize) etc.
(3) I ask the author to be careful about experimental description. No reader can perform their experiments in the present version. For example, line 117. What ratio did they mention (8:1)? Line 126, how did they evaluate the amount, 0.055 g of zeolite with 0.00688 g of nZVL? Line 166, there is no description about the conc about the bacteria.
(4) About the line 176-177, there is no 100% ethanol. And what is the difference between 90% and 100% ethanol?
(5) Line 201, What is the unit? (20 composite carriers??)
(6) Line 206, How did they measure the conc of ammonia nitrogen?
(7) Line 227, kF should be KF
(8) Line 252, What is this sentence? K is the correlation constant??
(9) Line 279-286. They did not define all the equation and this is very unkind to the readers: What is HRT? TN? eff and inf are also defined.
(10) Caption of Figure 1 is not adequate. They just observed the morphology. That is, they did not mention about (a), (b), and (c) in Figure1.
(11) In addition, there is no description in Figure 2(b)
(12) Line 338 and 341, please same style. Fe0 or Fe0
(13) Line 361, 362, it is not correct to use the word of charge. This is the zeta potential (mV)
(14) About Figure3,the figure size is too small to check them. So please re-arrange and enlarge them.
(15) Line 407 and 408, I could not understand what they described. Resist hydrophobicity?
(16) About Figure 4 (c), how did they evaluate?
(17) Line 427, I think Fig 5(b) is not correct but Fig 5(a) is correct. If this is correct, I could not find the description about Figure 5(b)
(18) About 5(b), which carrier did they obtain the data with?
Comments on the Quality of English LanguagePlease check my comments.
Author Response

(The authors gave the same response as above.)

Round 2
Reviewer 2 Report
Comments and Suggestions for Authors
The authors have substantially improved their manuscript, they responded to the comments one by one and in my opinion it can now be accepted.